# Resuscitative endovascular balloon occlusion of the aorta associated with improved survival in hemorrhagic shock

Melike N. Harfouche[1]*, Marta J. Madurska[2], Noha Elansary[1], Hossam Abdou[1], Eric Lang[1], Joseph J. DuBose[3], Rishi Kundi[1], David V. Feliciano[1], Thomas M. Scalea[1], Jonathan J. Morrison[1]

1 R Adams Cowley Shock Trauma Center, University of Maryland Medical System, Baltimore, Maryland, United States of America, 2 The Freeman Hospital, Newcastle upon Tyne, United Kingdom, 3 Dell Medical School, University of Texas at Austin, Austin, Texas, United States of America

* mharfouche@som.umaryland.edu

## Abstract

### Background

Resuscitative Endovascular Balloon Occlusion of the Aorta (REBOA) is controversial as a hemorrhage control adjunct due to lack of data with a suitable control group. We aimed to determine outcomes of trauma patients in shock undergoing REBOA versus no-REBOA.

### Methods

This single-center, retrospective, matched cohort study analyzed patients ≥16 years in hemorrhagic shock without cardiac arrest (2000–2019). REBOA (R; 2015–2019) patients were propensity matched 2:1 to historic (H; 2000–2012) and contemporary (C; 2013–2019) groups. In-hospital mortality and 30-day survival were analyzed using chi-squared and log rank testing, respectively.

### Results

A total of 102,481 patients were included (R = 57, C = 88,545, H = 13,879). Propensity scores were assigned using age, race, mechanism, lowest systolic blood pressure, lowest Glasgow Coma Score (GCS), and body region Abbreviated Injury Scale scores to generate matched groups (R = 57, C = 114, H = 114). In-hospital mortality was significantly lower in the REBOA group (19.3%) compared to the contemporary (35.1%; p = 0.024) and historic (44.7%; p = 0.001) groups. 30-day survival was significantly higher in the REBOA versus no-REBOA groups.

### Conclusion

In a high-volume center where its use is part of a coordinated hemorrhage control strategy, REBOA is associated with improved survival in patients with noncompressible torso hemorrhage.

**Data Availability Statement:** All relevant data are within the paper and its Supporting Information files.

**Funding:** The author(s) received no specific funding for this work.

**Competing interests:** I have read the journal's policy and the authors of this manuscript have the following competing interests: Jonathan J Morrison is on the Medical Advisory Board of Prytime Medical. The remainder of the authors have no conflicts of interest to report.

# Background

Resuscitative Endovascular Balloon Occlusion of Aorta (REBOA) has been gaining popularity over the past decade as an endoluminal adjunct to resuscitation in non- compressible sub-diaphragmatic torso hemorrhage (NCTH). Despite promising evidence that it can provide circulatory support in patients with hemorrhagic shock [1, 2], the use of REBOA has proven to be controversial as there is a lack of high quality evidence of clear survival benefit.

The current evidence base consists of large population studies using national trauma registries which demonstrate conflicting outcomes in terms of mortality of trauma patients treated by REBOA. A study by Norii and colleagues, using the Japanese trauma bank, utilized propensity score matching to compare trauma patients who received REBOA to those who did not and demonstrated that REBOA treatment results in mortality three times higher than controls [3]. In contrast, another group which used the same database but with a different propensity model demonstrated that severely injured patients treated with REBOA had a higher survival rate than those who did not receive REBOA [4]. Another study by a group that used the national American College of Surgeons Trauma Quality Improvement Program data set (ACS-TQIP) found that mortality was doubled in REBOA patients compared to no REBOA [5].

There is ongoing clinical uncertainty with regards to the use of REBOA in management of trauma patients. The evidence base is currently lacking data with a suitable control group from an experienced Level 1 Trauma Center. The aim of this study was to use the local trauma registry of one high-volume Level 1 Trauma Center to compare outcomes between trauma patients who were managed with REBOA and those who received standard treatment without REBOA.

# Methods

## Study population and data extraction

A retrospective review of the trauma registry at our institution was performed after obtaining University of Maryland Institutional Review Board approval. Request for waiver of documentation of informed consent was approved prior to study initiation. The trauma registry was developed for purposes of quality improvement and data monitoring and is a requirement for Level 1 trauma verification of our institution by the state of Maryland. It is a prospectively collected database that captures hundreds of variables ranging from demographic information to clinical presentation and outcomes. The findings are merged with databases from other trauma centers and used for national trauma outcomes reporting.

Patients were stratified into two groups: the REBOA group and the no-REBOA group. Within the no-REBOA group, historic (H = 2000–2012) and contemporary (C = 2013–2019) subgroups were created. The contemporary group was treated at a time when REBOA was available at our institution, whereas during the historic period it was not available. The use of historical controls was intended to mitigate selection bias, as unknown factors may have influenced the use of REBOA during the contemporary period. The rationale for a contemporary no-REBOA group was to control for bias associated with improvements in resuscitation and critical care management that would not have been available to the patients in the historic group. Although REBOA was being used in our institution as early as 2013, patients were included in the study starting in year 2015 to reduce poor outcomes being partially due to a learning curve after the device was initially introduced. In addition, more complete information regarding REBOA was available from 2015 onwards through the American Association for the Surgery of Trauma Aortic Occlusion for Resuscitation in Trauma and Acute Care Surgery (AORTA). Demographic characteristics as well as injury, physiology and outcome data

were collected from the registry. Variables not available in the trauma registry were identified through chart review, when available. Cause of death and laboratory values could not be obtained from the electronic medical record for the historic group. Specific information regarding indications, complications, and outcomes of REBOA was obtained from the local AORTA registry. Patients <16 years, as well as those in cardiac arrest upon arrival to the hospital were excluded. In addition, individuals missing data for any of the variables used to calculate the propensity score were excluded.

## Institutional setting

The R Adams Cowley Shock Trauma Center serves as a quaternary care center for the state of Maryland, functioning as an enhanced level 1 trauma center. The institution admits between 6000 and 7000 trauma patients annually. An endovascular trauma service staffed by trauma surgeons with vascular surgery training is available 24/7 to assist and support hemorrhage control endeavors [6]. REBOA use is governed by an institutional guideline and is a part of a well-coordinated hemorrhage control strategy. It is used primarily to bridge patients with NCTH or junctional hemorrhage to definitive hemorrhage control. The technique is performed only by highly trained operators who have been appropriately trained and certified in its use [7]. The device is deployed on average 3–5 times per month, or 30–60 times per year. There have been no changes to indications for REBOA placement during the study period.

## Data management and statistical analysis

Univariate analyses comparing demographic and clinical factors between the REBOA and no-REBOA groups (historic and contemporary) was performed using chi-square testing for categorical variables and the student's T-test for continuous variables. A logistic regression model was then used to assign a propensity score for each patient based on pre-treatment variables that were found to be significant on univariate comparison of REBOA to no-REBOA patients. These variables were: age, sex, race, mechanism of injury, injury severity score (ISS), lowest systolic blood pressure (SBP) and Glasgow Coma Score (GCS) within the first hour after arrival, and body region Abbreviated Injury Scale (AIS) score (brain, thorax, abdomen/pelvis and upper and lower extremities). Patients in the historic and contemporary groups were propensity matched 2:1 to the REBOA group (R = 2015–2019) using the nearest neighbor method to give the closest possible match in pre-specified criteria. A match tolerance of 0.001 was used. The Kaplan-Meier estimate was used to assess survival to 30 days in each group. Post-match univariate analyses were performed between the REBOA and no-REBOA groups for primary and secondary outcomes. The primary outcome of interest was in-hospital mortality. Secondary outcomes were 24-hour mortality, 30-day survival, length of stay, total blood products transfusion, acute kidney injury and lower limb complications. R statistical package version 3.0.1 was used for analysis and the Matchit package version 3.0.2 was used for the propensity scoring. P value of <0.05 was considered statistically significant.

## Results

A total of 130,651 patients were identified from the registry within the study period (H = 105,134, C = 25,410, R = 107). Patients were excluded due to age <16 (n = 2,518), arrival in cardiac arrest (n = 6,985) and incomplete data (n = 18,667). Incomplete data was missing at random, pertaining mostly to the Injury Severity Score and lowest SBP variables, and removal of these patients did not affect the overall averages of the variables used to calculate the propensity score in either no-REBOA group. Forty-eight patients were removed from the REBOA group due to arrival in cardiac arrest, and 2 had missing variables.

A total of 102,481 patients were included in the study (H = 88,545, C = 13,879, R = 57). Comparison of the REBOA group to the no-REBOA contemporary and historic groups by demographic, injury and physiology data is presented both pre- and post-match in Tables 1 and 2, respectively. Prior to matching, the REBOA group was significantly more likely to be male (R = 90% v C = 67% and H = 70%), have a higher body-region AIS and overall ISS (R = 34 v C = 10 and H = 11), lower systolic blood pressure (R = 67mmHg v C = 113 and H = 127) and lower GCS (R = 5 v C = 14 and H = 14) than the no-REBOA groups. When compared to the no-REBOA patients, the REBOA patients tended to be of younger age (R = 37y v C = 47y, H = 40y, p<0.001), and were more likely to have a penetrating mechanism (R = 23% v C = 13%, H = 13% p<0.001).

114 patients each in the contemporary and historic groups were matched to 57 REBOA patients. To determine if patients had been appropriately matched, baseline characteristics were compared. As demonstrated in Tables 1 and 2, patients in both the contemporary and historic groups did not differ in pre-treatment variables when compared to patients in the REBOA group after matching was complete. There were no differences in median levels of lactate (R = 6.1 vs C = 4.8, p = 0.073) or base deficit (R = 6.8 vs C = 7.6, p = 0.33) upon arrival between the REBOA and contemporary groups after matching.

In-hospital mortality was significantly lower in the REBOA group (19.3%) when compared to the contemporary (35.1%, p = 0.024) and historic (44.7%, p = 0.001) groups. Kaplan-Meier estimates of survival over time to 30 days demonstrated higher survival in the REBOA group compared to the historic (p = 0.035) and contemporary (p = 0.020) groups (Fig 1). Chi-square

**Table 1. REBOA to No-REBOA contemporary group before and after propensity-matching\*.**

| | Before Matching | | | After Matching | | |
|---|---|---|---|---|---|---|
| | No-REBOA (n = 13,879) | REBOA (n = 57) | *p* | No-REBOA (n = 114) | REBOA (n = 57) | *p* |
| Age, y | 47 ± 21 | 37 ± 14 | <0.001 | 42 ± 20 | 37 ± 14 | 0.194 |
| Sex n (%) | | | <0.001 | | | 0.050 |
| Male | 9326 (66.9%) | 51 (89.5%) | | 83 (72.8%) | 51 (89.5%) | |
| Female | 4607 (33.1%) | 6 (10.5%) | | 31 (27.2%) | 6 (10.5%) | |
| Race n (%) | | | <0.001 | | | 0.050 |
| White | 4814 (34.5%) | 24 (42.1%) | | 36 (31.6%) | 24 (42.1%) | |
| African-American | 7917 (56.8%) | 23 (40.4%) | | 72 (63.2%) | 23 (40.4%) | |
| Other | 1205 (8.6%) | 10 (17.5%) | | 6 (5.3%) | 10 (17.5%) | |
| Mechanism n (%) | | | <0.001 | | | 0.764 |
| Blunt | 11509 (80.8%) | 38 (66.7%) | | 81 (73%) | 38 (66.7%) | |
| Penetrating | 1806 (12.8%) | 13 (22.8%) | | 18 (16.2%) | 13 (22.8%) | |
| Other | 564 (6.4%) | 6 (10.6%) | | 12 (10.8%) | 6 (10.6%) | |
| Injury Severity Score | 10 ± 10 | 34 ± 15 | <0.001 | 38 ± 14 | 34 ± 15 | 0.420 |
| Lowest SBP, mmHg | 113 ± 22 | 67 ± 18 | <0.001 | 67 ± 21 | 67 ± 18 | 0.382 |
| Lowest GCS, mmHg | 14 ± 1 | 5 ± 3 | <0.001 | 4 ± 2 | 5 ± 3 | 0.399 |
| Body Region AIS | | | | | | |
| Brain | 1 ± 1 | 2 ± 2 | 0.003 | 2 ± 2 | 2 ± 2 | 0.100 |
| Thorax | 1 ± 1 | 2 ± 1 | <0.001 | 2 ± 1 | 2 ± 1 | 0.222 |
| Abdominal | 0 ± 1 | 3 ± 2 | <0.001 | 3 ± 2 | 3 ± 2 | 0.600 |
| Upper Extremity | 1 ± 1 | 1 ± 1 | <0.001 | 1 ± 1 | 1 ± 1 | 0.709 |
| Lower Extremity | 1 ± 1 | 2 ± 1 | <0.001 | 2 ± 1 | 2 ± 1 | 0.587 |

\*All values reported as median ± interquartile range unless otherwise stated.

**Table 2. REBOA to No-REBOA historic group before and after propensity-matching\*.**

| | Before Matching | | | After Matching | | |
|---|---|---|---|---|---|---|
| | No-REBOA (n = 88,545) | REBOA (n = 57) | *p* | No-REBOA (n = 114) | REBOA (n = 57) | *p* |
| Age, y | 40 ± 19 | 37 ± 14 | <0.001 | 38 ± 17 | 37 ± 14 | 0.969 |
| Sex n (%) | | | <0.001 | | | 0.050 |
| Male | 62,161 (70.2%) | 51 (89.5%) | | 80 (70.2%) | 51 (89.5%) | |
| Female | 26,367 (29.8%) | 6 (10.5%) | | 33 (28.9%) | 6 (10.5%) | |
| Unknown | 17 (0%) | 0 (0%) | | 1 (0.9%) | 0 (0%) | |
| Race n (%) | | | <0.001 | | | 0.313 |
| White | 52,352 (59.1%) | 24 (42.1%) | | 31 (27.2%) | 24 (42.1%) | |
| African-American | 29,746 (33.6%) | 23 (40.4%) | | 72 (63.2%) | 23 (40.4%) | |
| Other | 6,447 (7.3%) | 10 (17.5%) | | 11 (9.6%) | 10 (17.5%) | |
| Mechanism n (%) | | | <0.001 | | | 0.236 |
| Blunt | 71,166 (80.4%) | 38 (66.7%) | | 80 (70.2%) | 38 (66.7%) | |
| Penetrating | 11,380 (12.9%) | 13 (22.8%) | | 18 (15.8%) | 13 (22.8%) | |
| Unknown | 5,999 (6.6%) | 6 (10.6%) | | 16 (14.1%) | 6 (10.6%) | |
| Injury Severity Score | 11 ± 10 | 34 ± 15 | <0.001 | 33 ± 16 | 34 ± 15 | 0.553 |
| Lowest SBP, mmHg | 127 ± 18 | 67 ± 18 | <0.001 | 69 ± 21 | 67 ± 18 | 0.636 |
| Lowest GCS, mmHg | 14 ± 3 | 5 ± 3 | <0.001 | 4 ± 2 | 5 ± 3 | 0.479 |
| Body Region AIS | | | | | | |
| Brain | 0 ± 0 | 2 ± 2 | <0.001 | 2 ± 2 | 2 ± 2 | 0.589 |
| Thorax | 0 ± 0 | 2 ± 1 | <0.001 | 2 ± 2 | 2 ± 1 | 0.178 |
| Abdominal | 0 ± 0 | 3 ± 2 | <0.001 | 3 ± 2 | 3 ± 2 | 0.498 |
| Upper Extremity | 0 ± 0 | 1 ± 1 | <0.001 | 1 ± 1 | 1 ± 1 | 0.992 |
| Lower Extremity | 0 ± 0 | 2 ± 1 | <0.001 | 2 ± 1 | 2 ± 1 | 0.773 |

\*All values reported as median ± interquartile range unless otherwise stated.

comparison of mortality at 24 hours between the REBOA and no-REBOA historic group demonstrated lower mortality in the REBOA group (12% vs 28%, p = 0.014). There were no differences in 24-hour mortality when compared to the contemporary group (Table 3). Primary

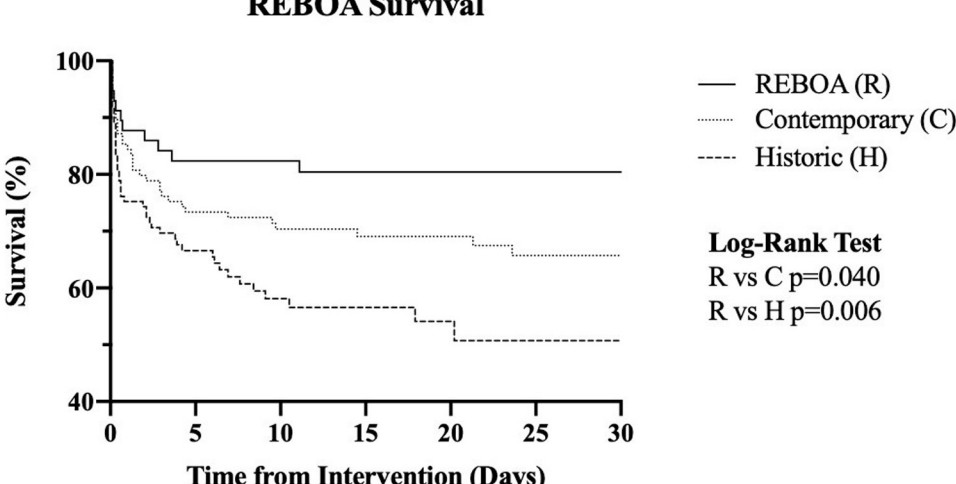

**Fig 1. Kaplan-Meier estimates of survival over time to 30 days by group (REBOA, contemporary and historic).**

**Table 3. Primary and secondary outcomes in REBOA and no-REBOA groups (contemporary and historic)[*].**

|  | REBOA | Contemporary | *p* | Historic | *p* |
|---|---|---|---|---|---|
| 24-hour mortality, n(%) | 7 (12.3%) | 22 (19.3%) | 0.175 | 32 (28.1%) | 0.014 |
| In-hospital mortality, n(%) | 11 (19.3) | 40 (35.1) | 0.024 | 51 (44.7) | 0.001 |
| 30-day mortality, n(%) | 4 (7%) | 18 (15.8%) | 0.081 | 19 (16.7%) | 0.062 |
| Total length of stay, d[a] | 29 ± 29 | 20 ± 20 | 0.030 | 9 ± 9 | < 0.001 |
| Total pRBC transfusions | 18 ± 18 | 19 ± 18 | 0.533 | 17 ± 14 | 0.498 |
| Acute Kidney Injury, n(%) | 13 (22.8%) | 28 (25%) | 0.455 | 27 (23.7%) | 0.530 |

[*]All values reported as median ± interquartile range unless otherwise stated, [a]Includes in-hospital deaths.

cause of death was mainly due hemorrhage in both the contemporary (n = 19, 48.7%) and REBOA (n = 7, 70%) groups, followed by traumatic brain injury (R = 1 [10%], C = 17 [43.6%]), and multifactorial (R = 2 [20%], C = 3 [7.7%]). Total length of stay was longer in the REBOA group by 20 days when compared to the historic group (p<0.001) and by 9 days when compared to the contemporary group (p = 0.03). There were no differences in acute kidney injury and total transfusions of packed red blood cells (pRBCs) between groups.

The overall incidence of lower extremity complications was low. A review of lower extremity complications in patients who underwent REBOA placement did not show any difference in rates of lower extremity amputation, exploration, fasciotomy or thrombectomy when compared to no-REBOA patients (Table 4).

When evaluating additional hemorrhage control procedures performed in each group, individuals in the REBOA group were more likely to undergo laparotomy than the contemporary or historic groups (79% vs 46.5% & 57%, respectively, p = 0.0003). Individuals in the contemporary group were more likely to undergo thoracotomy (C = 14.9% vs R = 7% & H = 2.6%, p = 0.004), and individuals in the historic group were more likely to undergo angiography (H = 29.8% vs C = 14% & R = 22.8%, p = 0.016). Amongst 53 patients for which data was available, zone of REBOA deployment was zone 1 for most individuals (N = 41, 77.4%) and zone 3 for the remainder (N = 12, 22.6%).

## Discussion

This is the first study from a high-volume trauma center in the U.S with considerable experience with REBOA that compares REBOA outcomes to a similar control group undergoing standard measures for hemorrhage control. Our findings demonstrate lower in-hospital mortality and improved 30-day survival in patients for which REBOA was used as compared to both a historical and contemporary cohort matched on injury severity, injury pattern and physiology. REBOA patients did not have increased acute kidney injury or lower extremity complications when compared to the no-REBOA groups. These findings underscore that REBOA is a valuable hemorrhage control tool that can reduce mortality when used in severe states of hemorrhagic shock.

**Table 4. Lower extremity complications in REBOA patients by lower extremity AIS score vs No-REBOA patients.**

|  | REBOA | Contemporary | p | Historic | p |
|---|---|---|---|---|---|
| Lower Extremity Amputation | 3 (5.3%) | 1 (0.9%) | 0.075 | 2 (1.8%) | 0.203 |
| Lower Extremity Exploration | 8 (14.0%) | 8 (7.1%) | 0.143 | 8 (7.0%) | 0.143 |
| Fasciotomy | 4 (7.0%) | 3 (2.6%) | 0.181 | 8 (7.1%) | 0.976 |
| Thrombectomy | 2 (3.5%) | 2 (1.8%) | 0.445 | 5 (4.4%) | 0.571 |

REBOA patients also experienced lower 24-hour mortality when compared to the historic group and demonstrated a trend towards reduced 24-hour mortality when compared to the contemporary patients which did not reach statistical significance. Concepts such as balanced blood product resuscitation, permissive hypotension and damage control surgery were newly entering practice during the historic period, which may have contributed to increased survival in both the REBOA and contemporary groups when compared to the historic group. Low overall numbers may have also influenced the non-significance of the comparison of 24-hour mortality between the REBOA and contemporary group. Greater blood transfusion requirements in the REBOA group are likely due to longer survival in these patients.

REBOA was originally described by Lieutenant Carl Hughes in 1954 as a method for controlling intra-abdominal hemorrhage during the Korean war [8]. However, due to limited availability of this device, it was not readily adopted at the time [9]. Since 2011, when it was reintroduced into clinical practice [10], its use has grown across trauma centers nationwide and its role in the management of NCTH has been met with both appraise [11, 12] and criticism [3, 5]. Despite the growth in utilization of the technique, there is a paucity of evidence evaluating REBOA from high-volume centers within the United States with an adequate control group. Much of the literature that exists to date are from international sites [4], national databases that include both low and high-volume centers [5] and national registries that do not provide a suitable control group, if any [13, 14].

The importance of evaluating REBOA outcomes in experienced centers cannot be overstated. A recent review of the American Association for the Surgery of Trauma (AAST) Aortic Occlusion for the Resuscitation in Trauma (AORTA) registry found that low-volume centers had a longer time to initiation of REBOA placement, longer time to aortic occlusion and lower odds of successful placement when compared to high-volume centers [15]. Critical to successful deployment of REBOA is early and expedient common femoral artery access [16], which can be challenging in a hypovolemic patient and is a technique that must be practiced regularly. Although REBOA volume by center has yet to be directly liked to clinical outcomes, the relationship between experience and performance has been demonstrated in several other procedural techniques. Given the introduction of the technique into the trauma landscape only 10 years prior, worldwide experience with REBOA is still building, and most centers are low-volume and still on the learning curve.

Recent reviews of REBOA have been conducted using large database analyses and/or in other countries, which has yielded results that are not highly applicable to high-volume centers in the United States. The study by Joseph et. al that demonstrated worse outcomes using REBOA used the Trauma Quality Improvement Program (TQIP) database from 2015–2016, which draws information from hundreds of Level I -III trauma centers across the US, many of which only recently started using REBOA [5]. Reports from the Japan Trauma Data Bank have been mixed regarding outcomes using REBOA, but their database includes a large rural population with trained ED providers deploying REBOA [3, 17]. Our institution is a Level I trauma center located in an urban setting with a high volume of penetrating trauma and high acuity blunt trauma, which is vastly different than the settings for REBOA use in Japan and other trauma centers in the US.

It is crucial that REBOA be a part of a coordinated hemorrhage-control strategy, whether that utilizes endovascular or open hemorrhage control techniques. At our institution, we provide 24/7 endovascular coverage by trauma-trained, vascular surgeons as part of an Endovascular Trauma Service which has resulted in faster times to hemorrhage control [6]. Similarly, we have a dedicated hybrid operating room for trauma which allows for rapid performance of concomitant endovascular and open procedures on patients who have undergone REBOA placement, if needed [18]. These resources ensure that REBOA is used in quick succession with other hemorrhage control techniques.

This study has some limitations that must be noted. Despite the superior ability of propensity matching to minimize bias and compare similar groups amongst highly heterogenous populations when compared to multivariable linear regression, it is still a retrospective, non-randomized analysis and can only determine associations rather than direct causation. It cannot control for unknown covariates that may influence the primary outcome, such as additional factors that affected the decision to place or not place a REBOA catheter, which may have resulted in selection bias. Another caveat of propensity matching is that all fields used for creating the propensity score must be filled. In this study, a high proportion of patients were removed due to missing data. This can unduly influence the results, as the characteristics of the study population are biased towards individuals that have all data available. By excluding patients who differ in their pre-treatment characteristics from the REBOA population, the findings demonstrate the average effect on the treated, which is a severely injured group in hemorrhagic shock, and not the entire study population. Hence, the results are only applicable to individuals with similar injury characteristics. Due to data limitations and the single center design, this is a small study that only includes 57 patients in the treatment group which should be taken into consideration when interpreting the results.

This study cannot determine the institution-specific factors that may have contributed to improved outcomes with REBOA, as these were not captured in the retrospective data. We can only speculate that high-volume REBOA users at our institution may have played a role in improving survival in the REBOA group. These same experienced surgeons also treated the patients who did not receive REBOA. The study results are applicable to centers that have a similar patient population, level of experience with REBOA, and resource availability to expediently manage subdiaphragmatic torso hemorrhage.

## Conclusion

This single-institution, propensity-matched, retrospective study comparing REBOA use to no-REBOA use in contemporary and historic cohorts demonstrated lower in-hospital mortality and improved 30-day survival for REBOA when compared to both contemporary and historic no-REBOA groups, and lower 24-hour mortality when compared to the historic group. Lower extremity complications were similar across groups. In a high-volume center where its use is part of a coordinated hemorrhage control strategy, REBOA is associated with improved survival in patients with noncompressible torso hemorrhage.

## Supporting information

**S1 Data.**
(XLSX)

## Author Contributions

**Conceptualization:** Melike N. Harfouche, Marta J. Madurska, Noha Elansary, Hossam Abdou, Eric Lang, Joseph J. DuBose, Rishi Kundi, David V. Feliciano, Thomas M. Scalea, Jonathan J. Morrison.

**Data curation:** Noha Elansary, Joseph J. DuBose.

**Formal analysis:** Melike N. Harfouche, Marta J. Madurska, Noha Elansary, Hossam Abdou, Eric Lang, Joseph J. DuBose, Rishi Kundi.

**Investigation:** Melike N. Harfouche, Marta J. Madurska, Noha Elansary, Hossam Abdou, Eric Lang, Joseph J. DuBose, Jonathan J. Morrison.

**Methodology:** Melike N. Harfouche, Noha Elansary, Hossam Abdou, Eric Lang, Joseph J. DuBose, David V. Feliciano, Jonathan J. Morrison.

**Project administration:** Jonathan J. Morrison.

**Resources:** Thomas M. Scalea, Jonathan J. Morrison.

**Supervision:** David V. Feliciano, Thomas M. Scalea.

**Writing – original draft:** Melike N. Harfouche, Marta J. Madurska, Jonathan J. Morrison.

**Writing – review & editing:** Melike N. Harfouche, Marta J. Madurska, Rishi Kundi, David V. Feliciano, Thomas M. Scalea, Jonathan J. Morrison.

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
