## [Decision Letter · Decision Letter 0]

14 Jan 2022

PONE-D-21-37832RESUSCITATIVE ENDOVASCULAR BALLOON OCCLUSION OF THE AORTA IMPROVES SURVIVAL IN HEMORRHAGIC SHOCKPLOS ONE

Dear Dr. Harfouche,

Thank you for submitting your manuscript to PLOS ONE. After careful consideration, we feel that it has merit but does not fully meet PLOS ONE’s publication criteria as it currently stands. Therefore, we invite you to submit a revised version of the manuscript that addresses the points raised during the review process.

We look forward to receiving your revised manuscript.

Kind regards,

Zsolt J. Balogh, MD, PhD, FRACS, FACS

Academic Editor

PLOS ONE

Journal Requirements:

Additional Editor Comments:

Dear Authors,

Your paper generated considerable interest among our senior reviewers. Unfortunately some of them found it quite marginal and some of them even recommended rejection.

My suggestion is to address all the concerns in your revision in an itemised fashion paying special attention to the details related to the actual level of evidence you are providing, the specific biases and the limitations of your study. Due to the nature of this work and the design the conclusions need to be more cautious and reserved.

I hope you consider the revision and provide us opportunity to re-review your work.

Best Regards,

Zsolt J. Balogh

Reviewers' comments:

Reviewer's Responses to Questions

**Comments to the Author**

1. Is the manuscript technically sound, and do the data support the conclusions?

Reviewer #1: Partly

Reviewer #2: No

Reviewer #3: Partly

Reviewer #4: Yes

2. Has the statistical analysis been performed appropriately and rigorously? 

Reviewer #1: Yes

Reviewer #2: No

Reviewer #3: I Don't Know

Reviewer #4: Yes

3. Have the authors made all data underlying the findings in their manuscript fully available?

Reviewer #1: No

Reviewer #2: Yes

Reviewer #3: Yes

Reviewer #4: Yes

4. Is the manuscript presented in an intelligible fashion and written in standard English?

Reviewer #1: Yes

Reviewer #2: Yes

Reviewer #3: Yes

Reviewer #4: Yes

5. Review Comments to the Author

Reviewer #1: 1. The authors state that REBOA improves survival in the title of the manuscript and throughout its body. This is a retrospective study, therefore the authors cannot state cause and effect, they can only report associations.

2. It appears the REBOA group consisted of patients treated during the period from 2015-2019 to allow a learning period after REBOA was started in 2011. The contemporary period was chosen between 2013 and 2019 and is not truly contemporary with relationhip to when REBOA was started or when the study of REBOA was started. Please explain the rationale for this choice. It seems the contemporary period should be the same as the period of study of REBOA.

3. The authors state that 60 REBOA catheters are placed in their institution per year but their study group only includes 57 patients who received REBOA over a 4 year period. Please explain.

4. The authors state in the limitations section that a large number of patients were removed from the study due to missing data needed for the propensity scoring. The data used for propensity scoring in this study are straight forward and should be available. Why are these data missing in so many patients? Does the removal of patients due to missing data bias the study? What types of patients were removed for missing data?

5. The authors state patient who received REBOA catheters were treated by highly experienced surgeons. Could the experience of the surgeon play a role in the improved outcome.

6. There are many factors that go into the decision of whether or not to place a REBOA catheter that are not captured in the propensity analysis. Patients may not have received a REBOA catheter because they were either too sick or not sick enough. These unknown factors not included in the propensity study could also bias the study. This should be discussed in the limitations.

7. After excluding all ineligible patients, only 57 patients who received a REBOA remain in this 4 year study. This is a small study. This should be discussed in the limitation section.

Reviewer #2: the authors are very experienced in the REBOA procedure and are leaders in this field. This work is an important contribution. Their stated aim, " The aim of this study was to use the local trauma registry of one high-volume Level 1 Trauma Center to compare outcomes between trauma patients who were managed with REBOA and those who received standard treatment without REBOA," is right on target.

However I find the inclusion of patients from the prereboa historical era distracting and not helpful at all. As they state many things have changed since that early time period. so much so that this groups should be dropped from the paper. Its simply distracting and not helpful. Additionally its not consistent with their stated aim.

The authors state that REBOA improves outcomes. However given that their data are retrospective, all they should say is that its associated with improved outcomes.

There is a trend towards decreased death in the reboa group vs contemporary. Please add a table on the causes of death in these patients.

Since reboa is supposed to decrease bleeding and hemorrhagic death, why wasn't there a decrease in transfusion and 24 hr deaths? Please address in the discussion.

Reviewer #3: Thank you for the opportunity to review this manuscript examining outcomes associated with the use of REBOA in a high-volume trauma centre.

The key limitation is the methodology. Although the authors have attempted to mitigate bias associated with a retrospective comparative study (and also discussed this in their limitations), the most significant issue is the likelihood of selection bias. This should be addressed further in the limitations discussion.

Also it would be helpful to understand what were the institutional indications for REBOA. Did these indications change over time?

Why were other markers of bleeding/shock not utilized in the propensity matching? ie heart rate, base deficit, lactate, MTP activation/number of PRBC transfused in the first 4 hours

Adding further variables would provide a more comparable control group

Why were the specific time periods selected for the dichotomy of “historic” vs “contemporary?” What changes in resuscitation/procedures/approaches occurred?

Although no direct comparison between the baseline characteristics of the historic vs contemporary cohorts was made, it appears that the contemporary group are older, less injured (by ISS, although I don’t understand why the AIS are higher in the contemporary group), and different in ethnicity. The outcomes appear to be improved in the contemporary cohort. Explaining the potential differences in the cohorts will assist the reader in understanding the findings of the study.

Reviewer #4: Nicely written work of relevance.

Main concern is around missing data which despite being discussed as limitation remains somewhat obscure. I suggest the authors provide more information in the methods and results section on what the missing important clinical data is rather than just generically stating this and only providing the total number in the results.

For completeness, the methods should also state that acute kidney injury is accounted for as a measured outcome given that it is.

Lastly, this : "60 times per year" and "but were available to the patients in the contemporary and REBOA groups" is redundant, the authors could remove it if they like.

6. PLOS authors have the option to publish the peer review history of their article (what does this mean?). If published, this will include your full peer review and any attached files.

Reviewer #1: No

Reviewer #2: No

Reviewer #3: No

Reviewer #4: No

---

## [Author Response · Author response to Decision Letter 0]

3 Feb 2022

February 3, 2022

Dear Dr. Balogh, 

Thank you to you and the reviewers for their thoughtful comments regarding our manuscript. Please find below a point-by-point response to each comment. A tracked version of the manuscript has been attached. We have uploaded the minimal anonymized data set necessary to replicate our findings as Supporting Information files. In addition, we have added a sentence describing study approval by the IRB. 

We hope you and the reviewers find the changes acceptable and the manuscript worthy of publication in PLOS ONE. 

Sincerely,

Melike Harfouche, MD

Corresponding Author

mharfouche@som.umaryland.edu

22 S Greene St, 

Baltimore, MD 21201

609-610-3165

Reviewer #1: 

1. The authors state that REBOA improves survival in the title of the manuscript and throughout its body. This is a retrospective study; therefore the authors cannot state cause and effect, they can only report associations.

Thank you for this comment. We have revised the title and body of the manuscript so that it states an association with survival rather than a cause and effect. 

2. It appears the REBOA group consisted of patients treated during the period from 2015-2019 to allow a learning period after REBOA was started in 2011. The contemporary period was chosen between 2013 and 2019 and is not truly contemporary with relationship to when REBOA was started or when the study of REBOA was started. Please explain the rationale for this choice. It seems the contemporary period should be the same as the period of study of REBOA.

Thank you for this question. We have clarified further our reasoning for this split in time periods. We wanted to select a contemporary time period during which REBOA was available to clinicians at our institution, where they could choose between using it/not using it and then select a time period when REBOA was not available, hence the patients were not “at risk” of being exposed to REBOA. That was the rationale for including historical controls, and for starting our contemporary period at the time REBOA was available to be used in 2013. We had incomplete data regarding patients who received REBOA between 2013-2015 as we were only newly entering these patients into the AORTA registry. The two-year delay in inclusion of REBOA patients allowed for minimization of bias from the learning curve associated with introducing a new technique. We have explained this further in the methods portion of the manuscript. 

3. The authors state that 60 REBOA catheters are placed in their institution per year but their study group only includes 57 patients who received REBOA over a 4 year period. Please explain.

Patients who received REBOA and were in cardiac arrest were excluded. Similarly, those with incomplete data were excluded as well. We originally identified 107 patients who were eligible for inclusion over the four years but had to exclude some for the aforementioned reasons. The range per year also varies from approximately 30-60 per year. We have adjusted this statement in the methods section so it is clearer. Thank you for pointing out this discrepancy. 

4. The authors state in the limitations section that a large number of patients were removed from the study due to missing data needed for the propensity scoring. The data used for propensity scoring in this study are straight forward and should be available. Why are these data missing in so many patients? Does the removal of patients due to missing data bias the study? What types of patients were removed for missing data?

Our trauma registry has undergone several changes over the years, which has led to the missingness of certain data points such as injury severity score, race and vital signs. These were the variables that were missing in most of the patients that were removed. However, removal of these patients did not shift the means of the variables included in the propensity score for either no-REBOA group. We have also added more detail regarding removal of the REBOA patients in the results paragraph. As these variables were missing at random, they did not bias the results of the study. 

5. The authors state patient who received REBOA catheters were treated by highly experienced surgeons. Could the experience of the surgeon play a role in the improved outcome.

Thank you for this comment. We indeed believe that surgeon experience may have played a role in the improved outcome in the REBOA group, but we cannot demonstrate this with the data. The same experienced surgeons treated the no REBOA patients. Please see the sentence in the last paragraph of the discussion which I have added onto the sentence starting with “We can only speculate…”. 

6. There are many factors that go into the decision of whether or not to place a REBOA catheter that are not captured in the propensity analysis. Patients may not have received a REBOA catheter because they were either too sick or not sick enough. These unknown factors not included in the propensity study could also bias the study. This should be discussed in the limitations.

Thank you. I have added to the sentence in the limitations section to make it clear that additional factors may have affected the decision to place or not place a REBOA catheter, which may have influenced the primary outcome of mortality. 

7. After excluding all ineligible patients, only 57 patients who received a REBOA remain in this 4 year study. This is a small study. This should be discussed in the limitation section.

I have added a sentence to the limitations section that emphasizes that this is a small study and should be interpreted as such. 

Reviewer #2: 

1. I find the inclusion of patients from the pre-REBOA historical era distracting and not helpful at all. As they state many things have changed since that early time period. so much so that this groups should be dropped from the paper. Its simply distracting and not helpful. Additionally its not consistent with their stated aim.

Thank you for these comments, and I hope we can address your concerns regarding including of a historical period. Our reasoning for including patients from 2000-2012 was to provide historic controls that were treated during a period when REBOA was not available. This would eliminate any bias associated with unknown factors that influenced whether REBOA was used during the contemporary period. However, due to changes in resuscitation techniques and advancements in trauma care, we also included a contemporary no-REBOA group. We have clarified this is in more detail in the methods section. 

2. The authors state that REBOA improves outcomes. However given that their data are retrospective, all they should say is that its associated with improved outcomes.

Thank you for pointing this out. We have adjusted the title and edited the discussion so it is clearer that this is an association and we cannot conclude a direct causation. 

3. There is a trend towards decreased death in the REBOA group vs contemporary. Please add a table on the causes of death in these patients.

Thank you for this point. We have done additional retrospective chart review to determine cause of death in both contemporary and REBOA groups. Hemorrhage was the primary cause of death in both groups, with a slightly higher proportion of deaths primarily attributable to TBI in the contemporary group which did not reach statistical significance. We have updated the results section with these findings. 

4. Since REBOA is supposed to decrease bleeding and hemorrhagic death, why wasn't there a decrease in transfusion and 24 hr deaths? Please address in the discussion.

Thank you for these comments. We have added a paragraph to the discussion regarding the lack of significance in the 24-hour survival of the REBOA vs contemporary group. We have also added a sentence explaining our thoughts regarding the lower transfusion requirements in the no-REBOA groups. 

Reviewer #3: Thank you for the opportunity to review this manuscript examining outcomes associated with the use of REBOA in a high-volume trauma centre.

1. The key limitation is the methodology. Although the authors have attempted to mitigate bias associated with a retrospective comparative study (and also discussed this in their limitations), the most significant issue is the likelihood of selection bias. This should be addressed further in the limitations discussion. 

Thank you. We have addressed this issue of selection bias in more detail in the limitations section. 

2. Also it would be helpful to understand what were the institutional indications for REBOA. Did these indications change over time?

Thank you for this comment. There were no changes to indications for REBOA placement during the study period. We have added two sentences to the Methods section under “Institutional Setting” to explain our approach to REBOA placement. 

3. Why were other markers of bleeding/shock not utilized in the propensity matching? ie heart rate, base deficit, lactate, MTP activation/number of PRBC transfused in the first 4 hours. Adding further variables would provide a more comparable control group.

Thank you for these comments. Unfortunately, base deficit, lactate, and MTP activation/blood transfusion in 4 hours are not variables that are available in the trauma registry which is why they could not be included in calculation of the propensity score. We strongly agree that they add additional information regarding the comparability of the groups. We have gone back and done an additional chart review to collect variables that are available in the electronic medical record and have added information regarding lactate and base deficit levels in the REBOA and contemporary groups post-match, confirming that they are in fact comparable. Please see changes in the results and methods section. We could not obtain this data for the historic group in the EMR. MTP activation and blood transfusion data cannot be reliably collected from the EMR retrospectively and hence we did not include them. We believe lowest systolic blood pressure within the first hour after arrival is more closely associated with mortality than initial heart rate, which is why we did not include it in calculation of our propensity score. 

4. Why were the specific time periods selected for the dichotomy of “historic” vs “contemporary?” What changes in resuscitation/procedures/approaches occurred?\\

Although no direct comparison between the baseline characteristics of the historic vs contemporary cohorts was made, it appears that the contemporary group are older, less injured (by ISS, although I don’t understand why the AIS are higher in the contemporary group), and different in ethnicity. The outcomes appear to be improved in the contemporary cohort. Explaining the potential differences in the cohorts will assist the reader in understanding the findings of the study.

Thank you for this question. We have added a more detailed explanation in the methods section regarding our reasoning for selecting historic (2000-2012) and contemporary (2013-2019) groups. The historic group is meant to represent a period when REBOA was not available, and hence reduce selection bias. However, changes in resuscitation and management techniques (primarily balanced resuscitation, damage control surgery and permissive hypotension) likely influenced outcome in the historic group, which is why a contemporary group was also included to compare to REBOA. This contemporary group was selected to start at the time REBOA was being used at our institution. We have added more to the discussion section to emphasize how changes in resuscitation techniques may have led to better outcomes in the REBOA and contemporary groups when compared to the historic group. 

Reviewer #4: Nicely written work of relevance.

1. Main concern is around missing data which despite being discussed as limitation remains somewhat obscure. I suggest the authors provide more information in the methods and results section on what the missing important clinical data is rather than just generically stating this and only providing the total number in the results.

Thank you for this question. The missing clinical data pertained to the variables used to calculate the propensity score (mainly injury severity score and vital signs). This was missing at random, and as such did not bias the study. The overall averages in the variables used to calculate the PS did not change after these patients were removed. We have added clarifying information to the methods and results section. 

2. For completeness, the methods should also state that acute kidney injury is accounted for as a measured outcome given that it is.

Thank you, this has been added. 

3. Lastly, this : "60 times per year" and "but were available to the patients in the contemporary and REBOA groups" is redundant, the authors could remove it if they like.

Thank you. We have removed the “were available…” portion of that sentence to make it less redundant. We have left the 60 times per year statement as it indicates that our center is considered high volume and may have influenced better outcomes.

---

## [Decision Letter · Decision Letter 1]

8 Mar 2022

RESUSCITATIVE ENDOVASCULAR BALLOON OCCLUSION OF THE AORTA ASSOCIATED WITH IMPROVED SURVIVAL IN HEMORRHAGIC SHOCK

PONE-D-21-37832R1

Dear Dr. Harfouche,

We’re pleased to inform you that your manuscript has been judged scientifically suitable for publication and will be formally accepted for publication once it meets all outstanding technical requirements.

Kind regards,

Zsolt J. Balogh, MD, PhD, FRACS

Academic Editor

PLOS ONE

Additional Editor Comments (optional):

thank you

Reviewers' comments:

Reviewer's Responses to Questions

**Comments to the Author**

1. If the authors have adequately addressed your comments raised in a previous round of review and you feel that this manuscript is now acceptable for publication, you may indicate that here to bypass the “Comments to the Author” section, enter your conflict of interest statement in the “Confidential to Editor” section, and submit your "Accept" recommendation.

Reviewer #1: (No Response)

Reviewer #2: All comments have been addressed

Reviewer #3: All comments have been addressed

Reviewer #4: (No Response)

2. Is the manuscript technically sound, and do the data support the conclusions?

Reviewer #1: No

Reviewer #2: Yes

Reviewer #3: Yes

Reviewer #4: Yes

3. Has the statistical analysis been performed appropriately and rigorously? 

Reviewer #1: Yes

Reviewer #2: Yes

Reviewer #3: Yes

Reviewer #4: Yes

4. Have the authors made all data underlying the findings in their manuscript fully available?

Reviewer #1: No

Reviewer #2: Yes

Reviewer #3: Yes

Reviewer #4: Yes

5. Is the manuscript presented in an intelligible fashion and written in standard English?

Reviewer #1: Yes

Reviewer #2: Yes

Reviewer #3: Yes

Reviewer #4: Yes

6. Review Comments to the Author

Reviewer #1: The authors have not substantially changed the manuscript in response to the reviewers comments. The inclusion of historical controls remains problematic due to the many changes that have occurred over time. It remains unclear why so many patients were excluded due to missing data when the data necessary for propensity matching should be present in any version of a trauma registry. The population studies is very small due to all of the exclusions but the conclusions are very bold in the face of the limitations.

Reviewer #2: The authors state the historical control groups is to mitigate selection bias. Since REBOA wasn't available during that time, selection isn't possible. I'm not sure why they use this group at all; selection bias is not a reason.

Reviewer #3: (No Response)

Reviewer #4: All comments adequately addressed except reviewer 2 comment 1, for which a supportive explanation was given. I think the review could be considered reasonably acceptable

7. PLOS authors have the option to publish the peer review history of their article (what does this mean?). If published, this will include your full peer review and any attached files.

Reviewer #1: No

Reviewer #2: No

Reviewer #3: No

Reviewer #4: No

---

## [Editor Report · Acceptance letter]

14 Mar 2022

PONE-D-21-37832R1 

Resuscitative Endovascular Balloon Occlusion of the Aorta ASSOCIATED WITH IMPROVED Survival in Hemorrhagic Shock 

Dear Dr. Harfouche:

I'm pleased to inform you that your manuscript has been deemed suitable for publication in PLOS ONE. Congratulations! Your manuscript is now with our production department. 

Kind regards, 

on behalf of

Dr. Zsolt J. Balogh 

Academic Editor

PLOS ONE